# Relation of Minimally Processed Foods and Ultra-Processed Foods with the Mediterranean Diet Score, Time-Related Meal Patterns and Waist Circumference: Results from a Cross-Sectional Study in University Students

**DOI:** 10.3390/ijerph20042806

**Published:** 2023-02-04

**Authors:** Paraskevi Detopoulou, Vassilios Dedes, Dimitra Syka, Konstantinos Tzirogiannis, Georgios I. Panoutsopoulos

**Affiliations:** 1Department of Clinical Nutrition, General Hospital Korgialenio Benakio, Athanassaki 2, 11526 Athens, Greece; 2Department of Nutritional Science and Dietetics, Faculty of Health Sciences, University of Peloponnese, New Building, Antikalamos, 24100 Kalamata, Greece; 3Internal Medicine Department, Mediterraneo Hospital, 16675 Athens, Greece

**Keywords:** ultra-processed foods, students, NOVA, Mediterranean diet, meal patterns

## Abstract

Ultra-processed foods are associated with chronic diseases, cardiometabolic factors and obesity. According to the NOVA system, foods are classified into four categories (from 1 = unprocessed to 4 = ultra-processed foods). The purpose of the present study was to assess the consumption of minimally processed foods (MPF) and ultra-processed foods (UPF) in university students and their relationship with obesity, Mediterranean diet adherence and meal patterns. In total, 346 students (269 women) of the University of Peloponnese participated. A food frequency questionnaire was used, and the MedDietScore was calculated. The % energy contribution of MPF and UPF was calculated. The identification of meal patterns was performed via principal component analysis. Both multivariate regression and Spearman’s correlations were used to measure the association of UPF/MPF consumption with anthropometric indices (body mass index, BMI and waist circumference, WC), Mediterranean diet adherence and early/late meal patterns. UPF and MPF provided 40.7 ± 13.6% and 44.3 ± 11.9% (mean ± standard deviation) of energy intake, respectively. In multi-adjusted linear regression models UPF consumption (% energy) was positively associated with WC in men but it was not related to BMI (total sample, men, women). UPF consumption was negatively related to the MedDietScore (Spearman rho = −0.214, *p* < 0.001) and an “early eating” pattern (Spearman rho = −0.120, *p* = 0.029) and positively associated with a “late eating” meal pattern (Spearman rho = 0.190, *p* = 0.001). MPF consumption was positively associated with the MedDietScore (Spearman rho = 0.309, *p* < 0.001) and an “early eating” pattern (Spearman rho = 0.240, *p* < 0.001). In conclusion, UPF consumption was positively related to WC in male university students. Nutritional and sociodemographic correlates of UPF consumption, such as low Mediterranean diet adherence and having a “late eating” pattern serve as a basis to better understand the UPF consumption-central obesity relation in young adults and should be considered in nutrition education programs for young adults.

## 1. Introduction

Ultra-processed foods (UPF) are defined as ready-to-eat, or ready-to-heat products, deriving from industrial procedures [1]. Examples of UPF are cereal bars, savory snacks, processed meats, pre-packaged frozen dishes, soft drinks, sweetened drinks, “instant” soups and sauces, etc. [1]. According to the degree of the procession, the NOVA food system classifies foods into four categories: category 1—unprocessed or minimally processed foods (MPF), category 2—processed culinary ingredients, category 3—processed foods (PF), and category 4—UPF [1]. Examples of foods belonging to NOVA categories 1–4 are the following: NOVA category 1 MPF: fruits, vegetables, nuts, eggs, etc.; NOVA category 2—processed culinary ingredients: olive oil, sugar, honey, butter; NOVA category 3—PF: tomato paste, freshly-made cheeses, canned fish, etc. NOVA category 4—UPF: fast foods, soft drinks, processed breads, biscuits, breakfast cereals and bars, sweetened yogurts, etc. Food processing affects food availability and food quality in a positive and negative way [2]. For example, several borne deriving diseases, such as hepatitis A or high microbial load, have been described for minimally processed foods (MPF) [3,4]. On the contrary, several health problems have also been attributed to UPF consumption [5], possibly related to their worse nutritional quality [6].

There is increasing evidence, according to meta-analyses, that the consumption of specific UPF, such as processed meat and beverages high in sugar, increases the risk of cardiovascular disease (CVD) [7], and mortality in cardiovascular patients [8]. The recent statement of the American Heart Association underlines that unprocessed forms of meat and poultry should be chosen and that MPF should be preferred over UPF to improve cardiovascular health [9]. Moreover, UPF consumption is related to CVD risk factors such as obesity, high blood pressure, and metabolic syndrome in epidemiologic studies [10,11,12]. Concerning obesity, many studies are reporting a positive relationship between UPF consumption and adiposity indices with a cross-sectional or longitudinal design [11,12,13,14,15], while there is only one clinical trial supporting the same hypothesis [16]. UPF are high in energy and saturated fat and they are palatable and convenient, which may, in part, explain their association with obesity [6].

This unique profile of UPF may also make them “attractive” food choices for university students for many reasons: new exposures, social circuits, food insecurity, poor cooking skills, migration, product characteristics, and an energy dense option if a main meal is skipped during classes and tight schedules [6,17,18,19,20,21,22,23]. Dietary habits can affect adiposity in university students [24]. More particularly, the embracement of healthy nutritional schemes, such as the Mediterranean diet [25,26,27], as well as meal characteristics, such as frequency, eating main meals, and breakfast [20,24,28,29], have been negatively associated with obesity in university students. Gender differences have also been described, with female students choosing more fruits, less fat and eating breakfast more frequently [30,31,32,33]. Data from epidemiological studies have shown that UPF consumption is related to younger age, male sex and obesity [34]. However, few studies have assessed the consumption of UPF in university students specifically [23,33,35,36,37], and even fewer their health-associated effects in this subgroup [36,37,38].

We have previously shown that dietary quality scores, such as the Food Compass Score, which includes NOVA categorization in its calculation, are associated with better nutritional habits and an “early eating” pattern in university students [39]. Moreover, a pattern high in fast foods and sweets, which are considered UPF, has been associated with a higher probability of overweight and obesity as well as a higher WC in men of the same cohort [24].

To our knowledge, there is no study investigating the levels of UPF consumption and specific UPF foods in Greek adults categorized with the NOVA system. Moreover, there are scarce data on the relation of UPF and health correlates of UPF in young adults [36,37,38] as well as chrononutrition, as expressed by meal/snack timing of intake [40]. Therefore, the rational of this study was to shed light in the UPF consumption in Greece, and its potential relation to adiposity and chronotype. More specifically, the present work aimed to assess the extent of UPF intake and investigate the relation of UPF and MPF consumption with adiposity and early/late meal patterns in a sample of Greek university students.

## 2. Materials and Methods

### 2.1. Study Design

From September to November 2018, a cross-sectional survey was conducted on students of the University of the Peloponnese, as previously described [39]. A “convenience sampling” approach was followed, mainly due to the fact that the University of the Peloponnese has departments in several towns. Students were recruited through advertisements at the university and central e-mail invitations. Moreover, volunteers that agreed to participate were instructed to also invite their colleagues. In total, three hundred and forty-six students participated in the study (269 women) from the following departments: “Nursing”, “Sports Organization and Management”, “Philology”, “History, Archaeology and Cultural Resources Management” and schools of Management, Agriculture and Food Technology of the Technical Educational Institution of Peloponnese, which was later incorporated in the University of Peloponnese (in 2019). The present sample constituted 8.6% of the registered students. The students were enrolled in several semesters of their studies. The University’s Ethics Committee (Faculty of Human Movement and Quality of Life Sciences) approved the rational of the study. All procedures were in accordance with the Declaration of Helsinki (1989) of the World Medical Association, as revised in 2013. All participants and the president of each department gave their informed consent.

### 2.2. Anthropometry

The detailed procedures followed in the study have been previously described [39]. Briefly, weight, height, WC and hip circumferences were measured. The Body Mass Index (BMI) was determined, and participants were classified as normal weight, overweight and obese [41]. Increased WC was determined according to the International Diabetes Federation cut-off values [42].

### 2.3. Meal Frequency and Early/Late Meal Patterns Assessment

Students stated how often they consume main meals and snacks [24]. In total, information was collected for the following meals: breakfast, morning snack, lunch, afternoon snack, dinner, and bedtime snack and several following frequencies. Then, scores were attached to meal frequency categories ranging from 1 (<once per week) to 5 (every day). Meal patterns in the present sample have been previously described [24]. Briefly, three time-related meal patterns were identified, i.e., “early eating” pattern, in which mostly consumed breakfast, morning snacks and afternoon snacks, “medium eating” pattern, in which lunch and dinner were consumed, and a “late eating” pattern, in which a bedtime snack was consumed.

### 2.4. Nutrition Assessment

A semi-quantitative food frequency questionnaire (FFQ) was used, as previously described [39]. The FFQ consisted of 156 foods/beverages and 9 frequency categories to choose from, from “never” to “everyday”. Food group intake was assessed [24], and the MedDietScore was used to evaluate Mediterranean Diet adherence [43]. The Schofield equation was used to estimate Basal metabolic rate (BMR) [44]. The assessment of underreporting was done by calculating the ratio of daily energy intake to BMR and using appropriate cut-off values (<1.09 for women and <1.07 for men) [45].

### 2.5. Ultra-Processed Foods Assessment

All foods included in the FFQ were assigned a NOVA categorization; (i) category 1 contains “unprocessed or minimally processed foods”. This means that it includes the edible plant or animal products that have been minimally modified/preserved; (ii) category 2 contains “culinary ingredients,” including salt, oil, sugar, or starch, (iii) category 3 contains “processed foods,” such as freshly baked bread, canned vegetables, or cured meats, which are obtained by categories 1 and 2 and (iv) category 4 contains “ultra-processed foods”, namely ready-to-eat products that contain additives. Then, the % energy contribution of MPF and UPF was calculated for each food. It is noted that the USDA database was used to determine the energy values of foods [46]. In total, 57 food items were considered to belong to the NOVA 4 category (for example, sodas, diet sodas, milkshakes, diet shakes, flavored yogurt, hot dog, sausages, ham, cereal bars, crackers, doughnuts, ice creams, mayonnaise, etc.). Moreover, 75 food items were considered to belong to the NOVA 1 category (for example, fresh fruits and vegetables, milk, rice, pasta, etc.).

### 2.6. Statistical Analysis

The Kolmogorov–Smirnoff test was used to test normality. Variables with a normal distribution are presented as means ± standard deviation (SD), while skewed distributed variables as medians and interquartile range. For categorical variables, absolute numbers and frequencies (%) are shown. Appropriate transformations were made to achieve normality, such as 1/WC and logarithmized BMI. For comparisons of normally distributed or transformed continuous variables between two groups, the *t*-test was applied, while for comparisons of skewed variables, the Mann–Whitney test was used. For comparisons of categorical variables between men and women, the Chi-square test was used and Bonferroni corrections were applied.

The identification of early/late meal patterns was performed with the use of Principal Components Analysis (PCA). The deriving scree plot was assessed. Moreover, the eigenvalues derived from the correlation matrix of the standardized variables were assessed and components having an eigenvalue more than 1 were kept for the data analyses. Three meal patterns were identified. Since component scores are considered as correlation coefficients (i.e., higher scores denote that the meal contributes most to the meal pattern), several meal patterns were defined about scores of variables that correlated most with the component (absolute loading value > 0.60). The orthogonal Varimax rotation was used to derive non-correlated patterns.

The association of UPF or MPF consumption (explanatory variables, expressed as % of total energy) with adiposity measures was tested with linear regression models. More particularly, logBMI and 1/WC were set as dependent variables, while adjustments were made for age, underreporting, department, living area (before enrollment) and gender (if applicable). In order to test the contribution of categories of UPF we applied linear regression models with each UPF category as an independent variable, as well as stepwise linear regression models. The variance inflation factor (VIF) was used to check for multicollinearity. A graphical check for potential heteroscedasticity was performed by plotting regression standardized residuals with 1/WC or logBMI for each variable entered in the model. Analyses were performed in the total sample and sex-specifically, given the previously reported sex differentiation in students’ diets in the present [24] and other samples [30,31].

Moreover, Spearman correlation coefficients were used to test the associations of UPF, MPF consumption and other variables, such as MedDietScore and early/late eating meal patterns derived from PCA analysis. *p*-values were based on two-sided tests. The significance level was set at 5%. SPSS Statistics for Windows was used for statistical analysis (version 22.0, Armonk, NY, USA, IBM Corp.). A posteriori power analysis was performed with the program G*power (version 3.1.9.7, Universität Kiel, Kiel, Germany).

## 3. Results

### 3.1. Basic Characteristics of the Volunteers

The primary characteristics of the subjects are presented in Table 1 and have been previously reported [24]. A higher participation rate was observed for the department of Nursing (34.7%). The number of female participants was higher than that of males (77.7% of the total sample were women). It is also noted that 44.2% of the total sample were first-year students. Women had a lower mean BMI and a lower rate of overweight than men. Moreover, women had a lower median value of WC than men but a higher incidence of increased WC according to pre-determined sex-specific cutoffs [42], i.e., 43.6% of women had WC higher than 80 cm and 19.1% of men had WC higher than 94 cm.

### 3.2. MedDietScore

The MedDietScore was 29 (24.5–32) for men and 30 (27.0–34.0) for women (medians, interquartile range, *p* < 0.01).

### 3.3. UPF and MPF Intake

The % energy contribution of UPF and MPF was calculated, i.e., the energy provided from foods in the NOVA categories 4 and 1, correspondingly. UPF provided 40.7 ± 13.6% of total energy (mean ± Standard deviation, SD) and no sex differences were documented. MPF provided 44.3 ± 11.9% of total energy (mean ± SD). MPF provided 44.3 ± 11.9% of total energy with no difference between men (44.0 ± 13.4%) and women (44.4 ± 11.5%) (means ± SD) (*p* = 0.826).

The energy from UPF food groups contributing more to the total were processed sweets, processed bread and fast-foods/pizza (Figure 1). In Table 2 the intakes of several UPF categories are shown in the total sample and for men and women, separately. As can be seen, men had a higher consumption of processed beverages (portions/day), ultra-processed meats (portions/day and % energy), sauces/condiments (portions/day) and fast foods/pizza (portions/day and % energy). Women had a higher consumption of processed cereals/bars and sweets (% energy) (all *ps* < 0.05). A positive association of UPF with energy intake was present in the total sample (Spearman rho = 0.372, *p* < 0.001), men (Spearman rho = 0. 307, *p* < 0.001) and women (Spearman rho = 0.372, *p* < 0.001).

It is noted that the median and interquartile range of energy from culinary ingredients and processed foods (PF) was 10.7% (6.4–14.1%) and 5.2% (3.0–8.0%), correspondingly. Women had a higher intake of energy from culinary ingredients (median, interquartile range: 11.2%, 6.7–14.4%) than men (median, interquartile range: 8.8%, 4.8–12.5%) (*p* = 0.014), while no sex difference was observed for PF between women (median, interquartile range: 5.1%, 3.0–7.8%) and men (median, interquartile range: 6.2%, 3.3–8.8%) (*p* = 0.140).

### 3.4. Relation of UPF and MPF Consumption with WC

In Table 3, linear regression models with 1/WC as dependent variable and UPF as the independent variable are displayed and adjusted for several co-variates. As can be seen, UPF consumption (% of energy) was positively related to WC in men, while no relation was documented in the total sample nor women. A graphical presentation of UPF and WC correlation for the total sample, men and women is shown in Appendix A. The inclusion of MedDietScore in the model led to a marginal statistical significance of UPF regarding WC in males (*p* = 0.059). The inclusion of the “early eater” pattern alone or in combination with the MedDietScore led to a loss of statistical significance in men (Appendix A). It is noted that in these models neither the MedDietScore nor the early/late eating patterns were significantly related to 1/WC.

Further models with individual foods as independent variable and 1/WC as dependent variable were applied in the total sample, men and women (Table 4). As it is seen, in these models no single food category was a predictor of 1/WC, with the exception of sweets in women, which were favorably related to WC. Stepwise linear regression models were also applied to test the same hypothesis (Table 5). In these models, ultra-processed beverages were negatively related to 1/WC in men. In women, ultra-processed dairy, and salty snacks were negatively related to 1/WC and sweets were paradoxically positively related to 1/WC.

The consumption of MPF (% of energy) was not associated with WC in the total sample and women in similar modes. It is noted that in men, MPF consumption was borderline negatively related to WC (b coefficient ± standard error of linear regression model with 1/WC as dependent variable: 3.4 × 10^−5^ ± 0.00001, *p* = 0.057, adj R-squared = 22.1%).

Culinary ingredients and PF were not associated with WC in the total sample and sex specific analysis (models not shown). In all models, VIF was 1.0–1.3 (less than 4), which suggests that there is no issue of multicollinearity. Furthermore, no issues of heteroscedasticity were detected.

### 3.5. Relation of UPF and MPF Consumption with logBMI

In Table 6, linear regression models with logBMI as dependent variable and UPF as the independent variable are displayed and adjusted for several co-variates. As can be seen, UPF consumption (% of energy) was not related to BMI in the total sample, as well as in men and women. Culinary ingredients and PF were not associated with BMI in the total sample and sex specific analysis (models not shown). In all models, VIF was 1.0–1.22 (less than 4), which suggests that there is no issue of multicollinearity. Furthermore, no issues of heteroscedasticity were detected. A graphical presentation of UPF and logBMI correlation for the total sample, men and women is shown in Appendix A.

### 3.6. Relation of UPF and MPF Consumption with MedDietScore

The Spearman correlation coefficients of early/late eating meal patterns with % total energy derived from UPF and MPF (for the total sample and both sexes) are presented in Table 7. As it is shown, the consumption of UPF and MPF was negatively and positively related to MedDietScore, respectively, in the total sample and women. A graphical presentation of UPF and MedDietScore correlation for the total sample, men and women is shown in Appendix A.

### 3.7. Relation of MPF and UPF Consumption with Early/Late Eating Meal Patterns

The Spearman correlation coefficients of early/late eating meal patterns with UPF and MPF consumption for the total sample and both sexes are presented in Table 7. As can be seen, UPF consumption was negatively related to the “early eating” pattern (total sample and men), while MPF consumption had the opposite association (total sample, men, and women). Moreover, UPF consumption was positively related to the “late eating” pattern (total sample and women). The “late eating” pattern was negatively associated with MPF consumption (total sample and women). A graphical presentation of UPF and meal patterns (early/late eating patterns) correlation for the total sample, men and women is shown in Appendix A.

## 4. Discussion

The rational of this study was to shed light on the UPF consumption in Greece, specific foods contributing to UPF intake and the potential relation of UPF to chronotype and health correlates, such as adiposity. The present study documented a positive association of UPF consumption with central adiposity in male university students, while there was no relation of UPF to BMI. Moreover, UPF consumption was negatively associated with adherence to the Mediterranean diet and an “early eating” pattern, and positively associated with a “late eating” pattern. MPF consumption was positively associated with Mediterranean diet adherence and an “early eating” pattern.

The present study adds to the existing literature, since few studies have addressed UPF consumption in young adults, such as university students [23,33,35,36,37] and even fewer have tested the effects of UPF consumption on health in this subgroup [36,37,38]. Moreover, there is no other study in a young adult Greek population assessing UPF consumption. Previous studies in Greece were performed on parents and showed that 32% of participants had a high intake of UPF [47], which is not directly comparable to our results, which are expressed as % of energy. Moreover, UPF provided 25.2% of total calories in a Greek menu for hospitalized subjects [48]. UPF provided 40.7 ± 13.6% of total energy in the present study. This is higher than other Mediterranean countries, such as Spain (24%) [49] and Italy (10%) [8], comparable to other countries such as France (36%) [34] and Australia (42%) [50], and lower than values reported for Britain (56.8%) [51], or the USA (58%) [52].

As recently reviewed, there are nine cross-sectional studies and seven longitudinal studies showing a positive association between UPF consumption and obesity [12]. The relation of UPF intake to obesity and/or central obesity can be explained by the fact that UPF constitute highly palatable, energy-dense food choices, which usually have higher sugar, sodium, and saturated fat content [6]. Moreover, UPF intake is related to increased sugar and decreased fiber consumption, as well as an increased eating rate, all of which constitute obesogenic factors [12,16]. They can also promote higher glucose peaks and decreased satiety compared to MPF [53]. In addition to nutrient-related mechanisms, UPF may have contaminants from food packaging, such as bisphenol-A (BPA) and phthalates, which have been linked to obesity [54,55]. Several additives used in UPF production, such as sodium benzoate, and monosodium glutamate, may also be obesogenic [55,56].

In the present study, UPF consumption was positively related to WC but not to BMI. Most available studies have assessed only BMI or the probability of being overweight and obese [12], while some have assessed WC and/or abdominal obesity [11,57,58,59]. It is of additive value that all prospective studies have shown a consistent increase in central adiposity with UPF consumption over time [60,61], a finding which is in the same direction with our results. However, it is noted that a positive relation of UPF consumption to both BMI and WC was observed in these studies [11,13,58], while in the study of Rauber et al. UPF consumption was positively related to odds of obesity but not to odds of abdominal obesity [51]. The apparent discrepancy of the above observations and our results may be explained by the relatively small sample size in our study and the population characteristics (such as younger age, Mediterranean population). Moreover, BMI is not as a good of a marker of adiposity as WC, since it may be increased in cases of individuals with increased muscle mass [62]. This phenomenon may be more prevalent in younger ages, since muscle mass is usually higher [62]. The inflammation-related correlations of BMI and abdominal fat may be also differentiated, as we have previously shown [63,64]. Interestingly, ultra-processed beverages have been related to increased values of WC, suggesting that additives, syrups and oligosaccharides, found in ultra-processed beverages can lead to insulin resistance and hormonal disturbances, relating to central and visceral adiposity [65,66].

There is only one clinical trial to investigate the causal role of UPF consumption in adiposity. In that randomized cross-over controlled trial in 20 weight-stable adults, the consumption of UPF for two weeks was related to an increased energy intake by ~500 kcal/day and weight gain of ~1 kg in the same time period [16]. Moreover, the study showed a reduction in ghrelin secretion, an increase in the satiety hormone PYY and a reduction in inflammatory markers with the unprocessed diet [16]. However, WC was not measured in this well-controlled study [16].

There are few studies examining sex differentiations in the relation of UPF consumption to obesity relationship, which show that the relationship is mostly apparent in women [57,58]. This is in contrast with our findings, since the UPF-central adiposity relation was documented in men in the present study and no relation was documented between UPF consumption and BMI. The observed discrepancy between our and the aforementioned studies is not related to the absolute intake of UPF, since the reported intake of UPF in the study of Pestoni et al. was 26% [57], in the study of Sung et al. was 57.5% [58] and in our study in between. However, the particularity of the Mediterranean population in the present study may in part explain the observed differences. For example, a recent study in Spanish students has shown that men had higher UPF consumption, and especially higher consumption of energy drinks, alcohol, and cereal bars, which could explain a possible association with obesity [35]. In the present study, men had a higher consumption of processed beverages, ultra-processed meats, sauces/condiments and fast foods/pizza, as portions/day and/or % energy, while women had a higher consumption of processed cereals/bars and sweets (% energy). Further studies are needed to clarify potential sex-differentiations in this diet-health hypothesis.

The inverse association of UPF consumption with Mediterranean diet adherence has been documented in the literature [67,68,69]. This association is in line with our findings and can be explained as a nutritional “shift” from fresh traditional Mediterranean meals to UPF, i.e., easy, ready-to-eat meals and beverages. It is noteworthy that the present study also found a relation of UPF consumption to a “late eating” pattern, which provides a sociological context for UPF consumption and a possible additional explanation of their association with obesity. Indeed, it seems that the participants who had a higher intake of UPF most commonly consumed snacks late. There are scarce data on the relation of UPF and chrononutrition, as expressed by meal/snack timing of intake [40]. A recent study found that subjects having an “early eating” (“morning”) pattern reported lower consumption of sweets, sweeteners, ultra-processed fats and seasonings [40]. Late meals have been associated with lower basal energy expenditure and obesity, as recently reviewed [70], underlying the emerging role of chrono-nutrition [70]. Mistimed eating can also result in hormonal dysregulation, such as leptin, cortisol, adiponectin, pro-opiomelanocortin, gastric inhibitory polypeptide and others, which can, in turn, promote weight gain [71]. Interestingly, the consumption of UPF has also been related to cancer [72] and cardiovascular risk factors [10], which are both related to obesity [73,74].

The strengths of our study include the assessment of UPF consumption in a Greek population not previously reported, as well as adjustment for underreporting. The following limitations should be taken into account. The cross-sectional design of our study cannot prove causality. Moreover, the generalizability of the presented results may be limited since university students in Peloponnese were included. The subgroup of men studied was relatively small. However, the achieved power with post-hoc power analysis taking into account the partial R^2^ corresponding to the explanatory ability of UPF consumption on 1/WC in men (R^2^ change = 0.152) was 97.3%. The clinical significance of our finding should be considered, since the beta coefficients in our model were relatively small. This, in part, may be explained by the multi-factorial nature of the measured dependent variables, i.e., WC and/or BMI. Moreover, most correlation coefficients between UPF, MPF consumption, MedDietScore and early/late eating meal patterns were <0.3, suggesting weak associations. A recent study has also reported a relatively low coefficient of association regarding Mediterranean diet score and UPF (rho = −0.35; *p* = 0.001) [67]. The magnitude of associations of dietary variables with early/late eating meal patterns is also in line with previous results from our group [24,39]. Several errors may be present in dietary assessment since an FFQ was used. However, we have partially accounted for potential estimation errors, since underreporting was considered in statistical analysis. The financial status of students, as well as their exact living conditions (alone, with roommates or family) were not assessed in the present study and these variables may affect UPF intake. Regarding the NOVA food classification system there is a debate about its usefulness since some argue that it has “little additive value” over nutrient values [75]. In contrast, others underline that ultra-processed foods also have a different physical structure [5]. Moreover, it is not always clear if some food products are UPF or not. For example, industrial bread made from simple materials such as wheat flour, water, salt and yeast, is a processed food, while if it contains emulsifiers or colors is UPF. These differentiations cannot be easily captured in nutritional epidemiology. Regarding meals, there is no consensus about the definition of snacks or meals [76]. Moreover, no information on the content/quality of meals consumed were available. Finally, data for energy expenditure were not available from the participants, which could potentially impact on our findings.

## 5. Conclusions

In conclusion, the present study confirmed a high UPF consumption among Greek university students, and provided the first evidence regarding UPF consumption in young Greek adults. The results of the present work showed that UPF consumption was positively related to WC in male university students. Nutritional and sociodemographic correlates of UPF consumption, such as low Mediterranean diet adherence and having a “late eating” pattern serve as a basis to better understand the UPF consumption-central obesity relation in young adults and should be considered in nutrition education programs when targeting reductions in UPF consumption to lower centripetal adiposity, associated with increased cardiometabolic risk.

## Figures and Tables

**Figure 1 ijerph-20-02806-f001:**
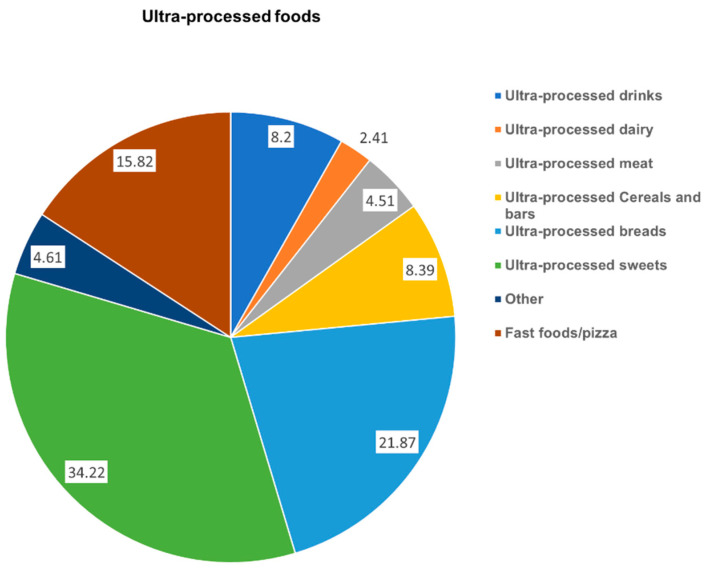
Daily energy intake from several categories of ultra-processed foods.

**Table 1 ijerph-20-02806-t001:** Descriptive characteristics of participants.

	Total *n =* 346	Men *n =* 87	Women *n =* 269	*p*
**Age (years)**	19.6 ± 3.1	19.3 ± 1.9	19.6 ± 3.4	0.278
**Year of study (%)**				
*1st*	44.2	50.6	42.3	0.162
*2nd*	16.2	29.8	12.2	<0.001
*3rd*	18.2	12.9	19.7	0.194
*4th*	20.8	5.19	25.2	<0.001
**Living area (before enrollment) (%)**				
*<50,000 habitats*	28.6	36.7	29.2	0.201
*>50,000 habitats*	71.4	73.3	70.8	0.322
**Department**				
*Nursing (%)*	34.7	28.5	36.4	0.201
*History, Archaeology and Cultural Resources Management (%)*	20.5	24.6	19.3	0.306
*Philology (%)*	25.4	22.0	26.3	0.443
*Sports Organization and Management (%)*	4.3	3.8	4.4	0.830
*Other (%)*	15	20.7	13.3	0.109
**ΒΜΙ (kg/m^2^) ** ^a^	22.0 (19.9–24.4)	23.7 (21.4–25.9)	21.6 (19.7–23.8)	0.002
*Normal weight (%) * ^b^	69	44.5	70.3	0.317
*Overweight (%) * ^b^	16.8	30	13	<0.001
*Obese (%) * ^b^	2.9	1.3	3.3	0.368
*Morbid obese (%) * ^b^	0.6	0	0.7	0.423
**WC (cm) ** ^a^	80.0 (74.0–89.0)	86.0 (79.0–90.7)	78.0 (72.0–89.0)	<0.001
*Increased WC (%)* ^c^	37.8	19.1	43.6	<0.001

**^a^** Transformations were applied to achieve normality (logBMI, 1/WC); ^b^ weight status categories were defined according to the criteria of the World Health Association [41]; ^c^ increased waist circumference was assessed by the International Diabetes Federation cut-offs [42]; means ± standard deviations for variables with normal distribution are shown. Medians and interquartile ranges (25th–75th) are shown for skewed variables. Categorical variables are displayed as absolute numbers and %. Student t-test was used to compare means of variables with a normal distribution. Mann–Whitney test was used to compare differences in non-normally distributed variables. Chi-square test was used to compare means of categorical variables. BMI: body mass index; WC: waist circumference.

**Table 2 ijerph-20-02806-t002:** Daily intake of UPF.

	Total *n =* 346	Men *n =* 87	Women *n =* 269	*p* †
Ultra-processed beverages (portions/day)	0.362 (0.124–0.796)	0.583 (0.138–0.985)	0.327 (0.114–0.712)	0.046
(% energy/day)	2.09 (0.81–4.65)	3.04 (1.22–5.66)	1.88 (0.77–4.01)	0.05
Ultra-processed dairy (portions/day)	0.083 (0.028–0.229)	0.111 (0.028–0.284)	0.083 (0.028–0.226)	0.659
(% energy/day)	0.2 (0.05–0.9)	0.47 (0.09–1.02)	0.26 (0.05–0.92)	0.251
Ultra-processed meats (portions/day)	0.116 (0.042–0.313)	0.203 (0.094–0.426)	0.097 (0.042–0.285)	<0.001
(% energy/day)	1.06 (0.36–2.3)	1.6 (0.70–3.3)	0.91 (0.32–2.06)	0.002
Ultra-processed cereals and bars (portions/day)	0.427 (0.082–1.016)	0.212 (0.044–1)	0.451 (0.114–1.02)	0.105
(% energy/day)	2.17 (0.50–4.7)	1.28 (0.24–4.5)	2.4 (0.64–4.8)	0.044
Ultra-processed bread (portions/day)	1.04 (0.494–1.72)	1.05 (0.607–1.803)	1.04 (0.485–1.69)	0.355
(% energy/day)	7.5 (4.4–11.8)	7.4 (4.5–11.4)	7.5 (4.3–11.9)	0.952
Sweets (portions/day)	1.21 (0.509–2.34)	1.10 (0.423–2.54)	1.26 (0.557–2.22)	0.711
(% energy/day)	10.8 (5.8–18.8)	9.1 (4.2–14.8)	11.1 (6.7–19.5)	0.007
Salty Snacks (portions/day)	0.033 (0.014–0.142)	0.033 (0.014–0.142)	0.033 (0.014–0.142)	0.674
(% energy/day)	0.41 (0.15–1.12)	0.35 (0.12–1.5)	0.41 (0.17–1.0)	0.500
Sauces (portions/day)	0.1965 (0.061–0.515)	0.263 (0.105–0.907)	0.184 (0.056–0.434)	0.022
(% energy/day)	0.38 (0.15–1.04)	0.18 (0.59–1.26)	0.35 (0.15–0.97)	0.075
Fast Foods/pizza (portions/day)	0.341 (0.1815–0.669)	0.415 (0.267–0.901)	0.318 (0.164–0.627)	0.005
(% energy/day)	5.0 (2.6–8.1)	6.5 (3.5–9.8)	4.6 (2.4–7.6)	0.005
Total UPF (% energy)	40.7 ± 13.6	41.0 ± 14.6	40.6 ± 13.3	0.845

Data are presented as median (lower–upper quartile) (25th–75th). For total UPF (% energy) mean ± standard deviation is shown. The Mann–Whitney test was used to compare total UPF intake and intakes of several food groups between men and women. The *t*-test was used to compare total UPF intake between men and women; † indicates *p*-value for comparing food portions or energy from food categories between men and women; one serving of sugary beverages was considered 250 mL of beverage. One serving of processed dairy was 15 g of cream, 200 g of flavored yogurt, 30 g of cream cheese. One serving of processed meats was considered as one sausage or 30 g ham. One serving of cereals/bars was considered as ½ cup of cereals or 1 bar. One serving of bread was considered 1 slice of bread, 2 melba toasts/crackers. One serving of sweets was considered as 5 biscuits (70 g), 1 waffle (75 g), 1 doughnut (43 g), a piece of cake (95 g), etc.

**Table 3 ijerph-20-02806-t003:** Linear regression analysis for l/WC in the total sample, men and women.

Dependent Variable: 1/WC	Total Sample adj R^2^ = 18.3%					Men adj R^2^ = 22.3%					Women adj R^2^ = 11.1%				
	B	SE	*p*	Tolerance	VIF	B	SE	*p*	Tolerance	VIF	B	SE	*p*	Tolerance	VIF
(Constant)	1.3 × 10^−2^	1 × 10^−3^	<0.001			1.5 × 10^−2^	3 × 10^−3^	<0.001			1.5 × 10^−2^	1 × 10^−2^	<0.001		
*Sex (men = 1, women = 0)*	1 × 10^−3^	3 × 10^−4^	0.009	0.777	1.287	NA	NA	NA	NA	NA	NA	NA	NA	NA	NA
*Age (y)*	−5 × 10^−5^	3 × 10^−5^	0.161	0.925	1.081	−6 × 10^−5^	9 × 10^−5^	0.555	0.889	1.125	−5 × 10^−5^	3 × 10^−5^	0.180	0.923	1.084
*Department (nursing = 1, others = 0)*	−1 × 10^−3^	2 × 10^−4^	0.003	0.964	1.037	−1 × 10^−3^	1 × 10^−3^	0.101	0.949	1.053	−1 × 10^−3^	3 × 10^−4^	0.010	0.963	1.038
*Living area (before enrollment) †*	7 × 10^−6^	2 × 10^−4^	0.870	0.982	1.019	2 × 10^−3^	4 × 10^−4^	0.001	0.956	1.046	−2 × 10^−4^	3 × 10^−4^	0.403	0.977	1.024
*Underreporting (yes = 1)*	−1 × 10^−3^	3 × 10^−4^	0.002	0.738	1.355	−1 × 10^−3^	4 × 10^−4^	0.100	0.875	1.142	−1 × 10^−3^	3 × 10^−4^	0.001	0.939	1.065
*UPF (% total energy)*	3.6 × 10^−6^	5 × 10^−6^	0.645	0.935	1.070	−3 × 10^−5^	1× 10^−5^	0.050	0.807	1.239	1.5 × 10^−5^	10^−5^	0.188	0.949	1.054

† 1 ≤50,000 habitats, 2 ≥50,000 habitats; B: unstandardized beta coefficient; SE: standard error; VIF: variance inflation factor; NA: not applicable.

**Table 4 ijerph-20-02806-t004:** Linear regression analysis for 1/WC in the total sample, men and women with processed food groups as independent variables (one group in each model).

Dependent Variable: 1/WC	Total Sample					Men					Women				
	B	SE	*p*	Tolerance	VIF	B	SE	*p*	Tolerance	VIF	B	SE	*p*	Tolerance	VIF
Ultra-processed beverages (% energy)	−5.1 × 10^−5^	3 × 10^−5^	0.127	0.978	1.023	−7.5 × 10^−5^	7 × 10^−5^	0.287	0.886	1.129	−5.6 × 10^−5^	4 × 10^−5^	0.144	0.961	1.040
Ultra-processed dairy (% energy)	−11 × 10^−4^	8 × 10^−5^	0.099	0.959	1.043	−3.7 10^−5^	1 ×10^−4^	0.779	0.915	1.093	−1.7 × 10^−4^	1 × 10^−4^	0.091	0.973	1.028
Ultra-processed meats (% energy)	2.5 × 10^−5^	5 ×10^−5^	0.703	0.936	1.068	1.2 × 10^−^4	1 × 10^−4^	0.275	0.890	1.123	−7 × 10^−6^	8 × 10^−5^	0.929	0.963	1.039
Ultra-processed cereals and bars (% energy)	6.3 × 10^−6^	3 × 10^−5^	0.853	0.992	1.008	6.2 × 10^−5^	6 × 10^−5^	0.332	0.924	1.082	7.4 × 10^−6^	4 × 10^−5^	0.854	0.977	1.024
Ultra-processed bread (% energy)	−2.6 × 10^−5^	2 × 10^−5^	0.213	0.984	1.016	−4.0 × 10^−5^	3.8 ×10^−5^	0.299	0.985	1.015	−2.1 × 10^−5^	2 × 10^−5^	0.385	0.979	1.021
Ultra-processed sweets (% energy)	2.7 × 10^−5^	1 × 10^−5^	0.026	0.921	1.085	−3.8 × 10^−5^	2 × 10^−5^	0.084	0.837	1.195	5.2 × 10^−5^	1 × 10^−5^	<0.001	0.931	1.075
Ultra-processed salty snacks (% energy)	−6.30 × 10^−5^	8 × 10^−5^	0.407	0.984	1.016	−1.5 × 10^−4^	2 × 10^−4^	0.475	0.801	1.248	−7.4 × 10^−5^	8 ×10^−5^	0.363	0.989	1.012
Sauces (% energy)	−2.19 × 10^−5^	9 × 10^−5^	0.806	0.975	1.026	−1.6 × 10^−4^	1.5 × 10^−4^	0.306	0.908	1.102	1.6 × 10^−5^	1 × 10^−4^	0.875	0.967	1.034
Fast foods/pizzas (% energy)	−6.67 × 10^−6^	2 × 10^−5^	0.775	0.944	1.059	−3.6 × 10^−6^	4 × 10^−5^	0.920	0.977	1.023	−1.0 × 10^−5^	3 × 10^−5^	0.732	0.944	1.059

All models are adjusted for age, sex (if applicable), department, living area and underreporting. To avoid multicollinearity one dietary variable per time was entered in each model. B: unstandardized beta coefficient; SE: standard error; VIF: variance inflation factor; NA: not applicable.

**Table 5 ijerph-20-02806-t005:** Linear regression analysis for 1/WC in the total sample, men and women with processed food groups as independent variables (stepwise linear regression model).

Dependent Variable: 1/WC	Total Sample ^1^ adj R^2^ = 10.6					Men ^2^ adj R^2^ = 26.7					Women ^3^ adj R^2^ = 21.0				
	B	SE	*p*	Tolerance	VIF	B	SE	*p*	Tolerance	VIF	B	SE	*p*	Tolerance	VIF
Ultra-processed beverages (% energy)	-	-	-	-	-	−9.33 × 10^−6^	3 × 10^−6^	0.006	0.877	1.140	-	-	-	-	-
Ultra-processed dairy (% energy)	-	-	-	-	-	-	-	-	-	-	−8.7 × 10^−6^	4 × 10^−6^	0.045	0.888	1.126
Ultra-processed meats (% energy)	-	-	-	-	-	-	-	-	-	-	-	-	-	-	-
Ultra-processed cereals and bars (% energy)	-	-	-	-	-	-	-	-	-	-	-	-	-	-	-
Ultra-processed bread (% energy)	-	-	-	-	-	-	-	-	-	-	-	-	-	-	-
Ultra-processed sweets (% energy)	-	-	-	-	-	-	-	-	-	-	3.0 × 10^−6^	5 × 10^−7^	<0.001	0.453	2.207
Ultra-processed salty snacks (% energy)	-	-	-	-	-	-	-	-	-	-	−1.23 × 10^−5^	3 × 10^−6^	<0.001	0.509	1.965
Sauces (% energy)	-	-	-	-	-	-	-	-	-	-	-	-	-	-	-
Fast foods/pizzas (% energy)	-	-	-	-	-	-	-	-	-	-	-	-	-	-	-

All models are adjusted for age, sex (if applicable), department, living area and underreporting. Only variables that were significant in the model are reported. -: Denotes that the variable was not a significant predictor of 1/WC and was not included in the final model. ^1^: In this model, only underreporting was a significant predictor. ^2^: In this model, underreporting, ultra-processed beverages, and department were significant predictors of 1/WC. ^3^: In this model, underreporting, ultra-processed dairy, sweets and salty snacks were significant predictors of 1/WC. B: unstandardized beta coefficient; SE: standard error; VIF: variance inflation factor.

**Table 6 ijerph-20-02806-t006:** Linear regression analysis for BMI (log) in the total sample, men and women.

Dependent Variable: LogBMI	Total Sample adj R^2^ = 14.7%					Men adj R^2^ = 3.4 %					Women adj R^2^ = 14.1%				
	B	SE	*p*	Tolerance	VIF	B	SE	*p*	Tolerance	VIF	B	SE	*p*	Tolerance	VIF
(Constant)	1.336	0.036	<0.001			1.359	0.090	<0.001			1.299	0.034	<0.001		
*Sex (men = 1, women = 0)*	−0.014	0.010	0.152	0.816	1.226	NA	NA	NA	NA	NA	NA	NA	0.047	NA	NA
*Age (y)*	0.003	0.001	0.022	0.933	1.071	0.004	0.004	0.359	0.921	1.086	0.002	0.001	0.288	0.927	1.079
*Department (nursing = 1, others = 0)*	0.001	0.008	0.925	0.982	1.018	−0.032	0.018	0.081	0.961	1.041	0.010	0.009	0.283	0.978	1.022
*Living area (before enrollment) †*	−0.011	0.007	0.159	0.973	1.028	−0.012	0.018	0.490	0.903	1.108	−0.009	0.008	<0.001	0.985	1.015
*Underreporting (yes = 1)*	0.045	0.009	<0.001	0.766	1.306	0.033	0.017	0.064	0.849	1.177	0.052	0.011	0.230	0.916	1.091
*UPF (% total energy)*	−2.4 × 10^−4^	2.7 × 10^−4^	0.381	0.925	1.081	−1.41 × 10^−6^	0.001	0.998	0.848	1.179	−3.8 × 10^−4^	3.2 × 10^−4^	0.171	0.930	1.075

† 1 ≤50,000 habitats, 2 ≥50,000 habitats; B: unstandardized beta coefficient; SE: standard error; VIF: variance inflation factor; NA: not applicable.

**Table 7 ijerph-20-02806-t007:** Spearman correlation coefficients between UPF, MPF consumption, MedDietScore and early, medium and late eating meal patterns.

		UPF (% Energy)	MPF (% Energy)
**MedDietScore (0–55)**	Total sample	−0.214 (*p <* 0.001)	0.309 (*p <* 0.001)
	Men	−0.089 (*p =* 0.440)	0.190 (*p =* 0.098)
	Women	−0.255 (*p <* 0.001)	0.339 (*p <* 0.001)
**Early eating pattern**	Total sample	−0.120 (*p =* 0.029)	0.240 (*p <* 0.001)
	Men	−0.230 (*p =* 0.05)	0.297 (*p =* 0.011)
	Women	0.086 (*p =* 0.169)	0.221 (*p <* 0.001)
**Medium eating pattern**	Total sample	0.010 (*p =* 0.858)	0.035 (*p =* 0.527)
	Men	−0.120 (*p =* 0.314)	0.160 (*p =* 0.176)
	Women	0.055 (*p =* 0.377)	−0.004 (*p =* 0.947)
**Late eating pattern**	Total sample	0.190 (*p <* 0.001)	−0.133 (*p =* 0.015)
	Men	0.034 (*p =* 0.776)	0.065 (*p =* 0.584)
	Women	0.234 (*p <* 0.001)	−0.187 (*p =* 0.002)

## Data Availability

Data are available upon request.

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
