# Peer review of "Relation of Minimally Processed Foods and Ultra-Processed Foods with the Mediterranean Diet Score, Time-Related Meal Patterns and Waist Circumference: Results from a Cross-Sectional Study in University Students"

_ijerph, 2023, doi:10.3390/ijerph20042806_

Round 1
Reviewer 1 Report
This manuscript is addressing through a new approach the contribution of ultra processed food in Mediterranean Diet Score, time-related meal patterns and waist circumference in students. The subject is interesting and the study design shows novelty. However the following issues with the manuscript structure should be addressed:
The novelty and the importance of the study should be more efficiently underlined in both the discussion section and conclusion as well as in the abstract. Why is this sample important? How helpful are these results?
In line 56, there should be a more clear explanation regarding the habit of skipping meals. Please rephrase.
In line 77 there is a reference of the study design description. Please include all relevant information there if some were omitted.
In line 149 it is stated: ''Moreover, women had a lower WC than men but a higher incidence of increased WC.'' Do you mean that although WC values for women are lower in numbers than men's, they have higher percentages above the desired limits? Because women's and men's values are not comparable . Please rephrase it in a better way or explain on why this comparison is important.
In general please check the use of English language especially in means of clarity of the presentation of the results.
Author Response
Reviewer #1
This manuscript is addressing through a new approach the contribution of ultra processed food in Mediterranean Diet Score, time-related meal patterns and waist circumference in students. The subject is interesting and the study design shows novelty. However the following issues with the manuscript structure should be addressed:
The novelty and the importance of the study should be more efficiently underlined in both the discussion section and conclusion as well as in the abstract. Why is this sample important? How helpful are these results?
We would like to thank the reviewer for his/her comments. We have updated the discussion and the conclusion of the manuscript regarding the importance and usefulness of the presented results.
In line 56, there should be a more clear explanation regarding the habit of skipping meals. Please rephrase.
Done
In line 77 there is a reference of the study design description. Please include all relevant information there if some were omitted.
More information has been added regarding the study design.
In line 149 it is stated: ''Moreover, women had a lower WC than men but a higher incidence of increased WC.'' Do you mean that although WC values for women are lower in numbers than men's, they have higher percentages above the desired limits? Because women's and men's values are not comparable. Please rephrase it in a better way or explain on why this comparison is important.
The authors have clarified this issue in the text. Women had a lower median value of WC than men but a higher incidence of increased WC according to pre-determined sex-specific cutoffs i.e., 43.6% of women had WC higher than 80 cm and 19.1% of men had WC higher than 94 cm. It is noted that increased waist circumference was determined according to the International Diabetes Federation cut-off values (methodology section).
In general please check the use of English language especially in means of clarity of the presentation of the results.
Several language errors have been corrected.
Reviewer 2 Report
This study has serious conceptual flaws as the results are well-established facts by definition. The Mediterranean diet is signified by the abundance of unprocessed food, hence the basic premise of the study has the fallacy that it aims to establish a fact that is established by definition.
Author Response
This study has serious conceptual flaws as the results are well-established facts by definition. The Mediterranean diet is signified by the abundance of unprocessed food, hence the basic premise of the study has the fallacy that it aims to establish a fact that is established by definition.
The authors agree that the Mediterranean diet is per definition an unprocessed dietary pattern and thus it would be automatically inversely related to UPF consumption. However, the novelty of our study is based on different features. Indeed, the present study adds to the existing literature since few studies have addressed UPF consumption in young adults, such as university students [1–5] and even fewer have tested the effects of UPF consumption on health in this subgroup [4–6]. Moreover, there is no other study in a young adult Greek population assessing UPF consumption. The focus on young students is of great importance since previous data point that UPF consumption is related to younger age [7] and, on top, university students may have difficulty in identifying healthy products from food labels [3]. In this context, the quantification of UPF in this subpopulation, the identification of frequently consumed ultra-processed products and their potential relation to abdominal adiposity could serve as a basis to better understand the UPF-obesity relation in young adults and be able to modify it.
- Dezanetti, T.; Quinaud, R.T.; Caraher, M.; Jomori, M.M. Meal Preparation and Consumption before and during the COVID-19 Pandemic: The Relationship with Cooking Skills of Brazilian University Students. Appetite 2022, 175, 106036, doi:10.1016/j.appet.2022.106036.
- Sprake, E.F.; Russell, J.M.; Cecil, J.E.; Cooper, R.J.; Grabowski, P.; Pourshahidi, L.K.; Barker, M.E. Dietary Patterns of University Students in the UK: A Cross-Sectional Study. Nutr J 2018, 17, 90, doi:10.1186/s12937-018-0398-y.
- Fondevila-Gascón, J.-F.; Berbel-Giménez, G.; Vidal-Portés, E.; Hurtado-Galarza, K. Ultra-Processed Foods in University Students: Implementing Nutri-Score to Make Healthy Choices. Healthcare 2022, 10, 984, doi:10.3390/healthcare10060984.
- Durán-Agüero, S.; Valdés-Badilla, P.; Valladares, M.; Espinoza, V.; Mena, F.; Oñate, G.; Fernandez, M.; Godoy-Cumillaf, A.; Crovetto, M. Consumption of Ultra-Processed Food and Its Association with Obesity in Chilean University Students: A Multi-Center Study: Ultra-Processed Food and Obesity in Chilean University Students. Journal of American College Health 2021, 1–7, doi:10.1080/07448481.2021.1967960.
- Santana, J. da M.; Milagres, M.P.; Silva dos Santos, C.; Brazil, J.M.; Lima, E.R.; Pereira, M. Dietary Intake of University Students during COVID-19 Social Distancing in the Northeast of Brazil and Associated Factors. Appetite 2021, 162, 105172, doi:10.1016/j.appet.2021.105172.
- Yarin-Achachagua, A.J.; Soria-Villanueva, L.M.; Tejada-Mendoza, M.A.; Arista-Huaco, M.J. Physical Condition and Eating Habits in Physical Education Students. In Proceedings of the Journal of Human Sport and Exercise - 2021 - Winter Conferences of Sports Science; Universidad de Alicante, 2021.
- Julia, C.; Martinez, L.; Allès, B.; Touvier, M.; Hercberg, S.; Méjean, C.; Kesse-Guyot, E. Contribution of Ultra-Processed Foods in the Diet of Adults from the French NutriNet-Santé Study. Public Health Nutr. 2018, 21, 27–37, doi:10.1017/S1368980017001367.
Reviewer 3 Report
This study examine the associations between the degree of the procession of food and obesity, Mediterranean diet adherence and meal patterns in university students. This study is interesting and gives us insights of adverse effects of ultra-processed foods, especially in young population.
However, there are a few concerns.
Measure points
I'm not sure if authors confirmed assumption of the liner regression analysis. Could you report the test of multicollinearity or homoscedasticity?
I also wonder why authors did not use Category 2 (Processed culinary ingredients) and Category 3 (Processed foods) for liner regression analysis.
I think they can perform liner regression analysis with BMI as dependent variables and each food categories as independent variables to see how much each category contributes to BMI or some other variables.
I wonder if these students lived alone or with roommates or family? I also wonder if authors measure students' household income. These factors would be associated with their diets. So if authors measured these factors, they should be included into the liner regression analysis.
Minor points
The authors should spell out these abbreviations when they appeared first.
NOVA, CVD, and BMR
Reviewer 4 Report
The study describes the association of ultra-processed food consumption with anthropometric measures and Mediterranean diet adherence and early/late meal consumption patterns among sample of the students of University of the Peloponnese in Greece (87 men, 269 women). The study has showed the UPF consumption was directly related to waist circumference in male students. It was also associated with low Mediterranean diet adherence and late eating pattern, particularly among female students.
This descriptive study provides valuable information and covers an interesting topic, but has some shortcomings:
General comments
· Abstract, Line 23: “…UPF were negatively associated with 1/waist circumference” – Why 1/waist circumference was used for regression analysis? Is it more logical to use waist circumference (WC) instead of 1/WC in regression analysis, and therefore it should be: “…UPF were positively associated with waist circumference”, which is easier, more logical, and straightforward for understanding. Additionally, in methods, there was not given an explanation for such strange way of analyzing results, with “1/WC” instead of WC. In contrast, in methods, BMI was stated to be used normally, not as “1/BMI”. Since there is no scientific explanation for usage of 1/WC in the analyses, analyses should be repeated with WC, not 1/WC. The same for table 3, recalculate coefficients with WC, not 1/WC, since it is more logical.
· Why in the Results there are not data presented for BMI, as indicated in Methods? Please, add a regression table also for BMI.
· Abstract: Why somewhere linear regression coefficients were presented, and somewhere rho coefficients, while only PCA was mentioned in the abstract for analyses? Therefore, in the abstract should be clearly stated that both multivariate regression and Spearman’s correlations were used to measure the association of UPF/MPF with anthropometric measures (BMI and WC), and the Mediterranean diet and early/late meal patterns.
· Table 2: it would be more correct to present % of total energy intake instead of energy in calories for all food groups , because it is known that men consume more energy than women, so the deference in consumption expressed in absolute values of energy (“energy/day”) is not relevant. The data in Table 2 should be expressed as % of total daily energy intake per person (“% energy/day” , as in figure 1), and % of energy derived from specific UPFs compared between men and women.
· In methods, it was not specified in linear regressions and Spearman’s correlation which data were used for UPF/MPF: “portions/day” or “energy/day”. It should be “% energy/day”.
· In methods, there is no mention of Spearman’s correlation, why?
· Why late eating pattern and the Mediterranean diet score were not also added as predictors in the regression model for WC? Maybe, an additional regression model (model 2) should be made, to see if the association of UPF with WC in men will be then lost, or even more pronounced.
· The majority of discussion is on association of UPF with BMI and obesity, while actually this study did not find the association with BMI/obesity (data not presented). The emphasis of discussion should be on the lack of association with BMI, and only association with WC in men.
· No study power or sample size calculation.
Small corrections:
Table 1: decimal places should be set uniformly to 1 decimal place (e.g., for %, age, BMI, WC), except for p values (there should be 3 decimal places uniformly)- now, somewhere there is no decimal place, somewhere there are 2 decimal places, somewhere there is 1 decimal place.
The same for table 2, unless the values are lower than 0 (e.g., for (portions/day), it can stay on 2 or 3 decimal places, and for p values (set to 3 decimal places uniformly).
For table 3, express B and SE uniformly e.g., “5 x 10-5”, “5 x 10-2”, not “0.05” (for p values, set to 3 decimal places uniformly).
Line 21: instead of “Meal patterns were analyzed via principal component analysis” it should be “The identification of UPF and MPF patterns was performed via principal component analysis”. (more precise and correct)
Line 38: “(UPF)” should be without brackets “UPF”
Lines 99-101: “The assessment of underreporting was done by calculating the ratio of daily energy intake to BMR and using appropriate cut-off values”- Please, add here also information how BMR was calculated for each individual in this study, according to which formula?
Line 104-105: it would be more correct to say “edible plant or animal products that have been minimally modified/preserved” instead of ” edible plants or animals that have been minimally modified/preserved”, since milk, eggs are animal products, not animals (refers only to meat/fish).
Lines 134-148: why it was not mentioned in the methods that the Spearman correlation coefficients were used to assess the associations of Mediterranean diet and early/late meal patterns with UPF and MPF? Please, add this information. Also specify in Methods what was used for linear regressions or Spearman correlations: UPF/MPF “portions/day”, “energy/day” or “% energy/day”. It should be “% energy/day”.
Why there is no the Results section on linear regression with BMI? In methods it was written: “More particularly, BMI and 1/WC were set as dependent variables while adjustments were made for age, underreporting, department, living area (before enrollment) and gender (if applicable).” So, please, add a regression table for BMI.
In Table 3, B coefficient for age has a positive value for total sample (5 x 10-5), while for women and men separately, it has negative values (as expected). I suppose that it should be a negative number also for total, isn’t it? Anyhow, please, make tables with WC as dependent variable, not 1/WC.
What * means in table 3?
Why late eating pattern and the Mediterranean diet score were not added as predictors in regression model for WC/BMI? Please, made an additional model, with those 2 variables also included (it can be a supplementary table).
Line 214: Section “3.5. Relation of UPF and MPF with MedDietScore” - It should be Section 3.6.: “3.6. Relation of UPF and MPF with MedDietScore”– why it is not more emphasized in the manuscript text that the associations were significant only in women?
Line 219: Sections 3.6. and 3.7. should be merged into one section “3.7. Relation of UPF and MPF consumption with early/late meal patterns”.
Line 224: Instead of “3.7. Relation of minimally and ultra- processed with meal patterns” it should be “3.7. Relation of UPF and MPF consumption with early/late meal patterns”.
The correction “early/late meal patterns” (more specific) should be made everywhere in the text where applicable.
Line 225-226: “The Spearman correlation coefficients of meal patterns with UPF and MPF for the total sample and both sexes are presented in Table 4” should be corrected into: “The Spearman correlation coefficients of early/late meal patterns with % total energy derived from UPF and MPF (for the total sample and both sexes) are presented in Table 4.”
Also mention significant differences between sexes in the text. The correlations coefficients are very different for men and women.
Line 242: “and obese” should be “and obesity”
Line 243: “of being overweight and obesity” should be “of being overweight and obese”
In discussion, the results of no association of UPF/MPF with BMI in the present study should be compared with the results of the studies which have showed an association. Is there any explanation why only associations with WC and only in men were found?
Differences between men and women should be discussed after comparing specific UPF food subgroups in terms of % of total energy intake, not in absolute values.
Were the late eating pattern and the low Mediterranean diet adherence associated with WC and BMI, both in men and women (did the authors perform a Spearman's correlation for these associations? It should be added)? Why the late eating pattern and the Mediterranean diet score were not added in an additional regression model for WC/BMI?
Maybe additional linear regression models with specific UPF groups can be made, to see which UPF subgroup contributes the most to WC?
Study limitations: small number of men included (n=87) does not allow adequate study power for analyses in this subgroup. Additionally, there is no sample size calculation. Please, add these limitations.
Even though it was stated that Bonferroni corrections were applied, I do not see where is was applied, in which tests? Please, underline those P values where the Bonferroni correction was applied.
- Is the manuscript clear, relevant for the field and presented in a well-structured manner? -Yes
- Are the cited references mostly recent publications (within the last 5 years) and relevant? -Yes
- Does it include an excessive number of self-citations? -No
- Is the manuscript scientifically sound and is the experimental design appropriate to test the hypothesis? – It must be improved (see above)
- Are the manuscript’s results reproducible based on the details given in the methods section? -More/less
- Are the figures/tables/images/schemes appropriate? Do they properly show the data? Are they easy to interpret and understand? Is the data interpreted appropriately and consistently throughout the manuscript? Please include details regarding the statistical analysis or data acquired from specific databases. – This part must be improved: some data are missing, additional testing should be performed (see above)
- Are the conclusions consistent with the evidence and arguments presented? -Needs to be more precise
- Please evaluate the ethics statements and data availability statements to ensure they are adequate. -Adequate
Round 2
Reviewer 2 Report
Specific comments:
-
The unstandardized beta coefficient reported in table 3 is in the range of 10-2 to 10-6. Please consult a statistician to confirm whether interpretations can be drawn using such small beta coefficients.
-
Line 213 - 214: “As can be seen, 213 UPF consumption (% of energy) was positively related to WC in men”. What about “total” and among “women”?
-
Line 227 - 228: “As can be seen, 227 UPF consumption (% of energy) was not related to BMI”. What about “total” and among “women”? How do authors explain, the correlation with WC but not with BMI?
-
The Spearman correlation coefficient in all comparisons shown in table 6 is below 0.3 (except in 2 instances), showing a very weak correlation.
-
Kindly present graphs for the main conclusions of the papers, since the correlations co-efficient are so small, visual confirmation of any correlation would make it easier to interpret, namely: UPF vs WC, UPF vs BMI, UPF vs Med diet score, UPF vs early eating pattern, UPF vs late eating pattern, for all three groups (Total, men, and women).
-
Have you quantified the correlation between UPF intake and total calorie intake?
- The discussion can be made more condensed and focused for better understanding.
Reviewer 3 Report
The authors responded to all my comments. There are no further comments and I think this revised manuscript is suitable for publication.
Author Response
We would like to thank the reviewer for the comments.
Reviewer 4 Report
The authors made required corrections and significantly improved their manuscript.
Just several more corrections will be needed:
I am not certain why for BMI log transformation was performed (the usual method for normalization), and not also for WC? Did logarithmic transformation for WC not result in normality of data?
Tables: It is unclear now for tables 4 and 5 what is presented in the tables. Please, accept the changes in the word document for tables (it is confusing like this is now with deletions and insertions, please download the PDF file) and arrange parts of tables 4 and 5 properly. (Please, also check if all other tables are there and properly numerated)
“Table 5. Linear regression analysis for 1/WC in the total sample, men and women with processed food groups as independent variables (one group in each model).”
Is in the regression analysis for % energy of 9 specific UPF groups (Table 5) omitted % energy of total UPF? It should be, but now it looks in the model there is still % energy of UPF.
In the regression model 2 (specific UPF), instead of % energy of total UPS, should be entered all 9 specific UPF groups in the model, and then the stepwise regression performed, which will eliminate those which are insignificant and keep only those which are associated with 1/WC. Like this, better visibility of those which are associated with WC will be acheived. Alternatively, since VIFs are adequate for the “enter” method (all variables included in the model), this method can be performed also, but % energy of total UPF should be omitted, and all specific UPF groups should be included in the method together. It is written: “Table 5. Linear regression analysis for 1/WC in the total sample, men and women with processed food groups as independent variables (one group in each model).”- which is quite confusing. All 9 UPF groups should be in the model together (without total UPF), and then “enter” or “stepwise” regression performed.
However, I suppose that the “enter” method will diminish significance of specific UPF groups (particularly after the Bonferroni correction of p value, which will be now really needed to be applied). Probably is worth testing both methods for regression (stepwise and enter), to see which will give better results. Now, it seems that eating sweets is negatively associated with WC, particularly in women, which is kind of strange finding.
(I suppose that all other variables/confounders included in the model, including sex, age, department, living area and underreporting were tested if the satisfy all presumptions for linear regression, isn’t it? )
“The inclusion of MedDietScore in the model led to a marginal statistical significance of UPF regarding WC in males (p=0.059). The inclusion of the “early eater” pattern alone or in combination with the MedDietScore led to a loss of statistical significance in men. We have added this point at the results.” (author’s reply file) - I have not seen this section in the results. In which section it was mentioned? Can you make these tables as supplementary tables? Additionally, why the late eating pattern was not included, it is expected to be more associated with WC? Actually, the scale early/medium/late pattern should be entered in the model (coded early= 1/medium=2/late=3).
Abstract, Line 26-27: “were positively associated with WC in men (with regression model with 1/WC as dependent variable)” – I would suggest to eliminate “(with regression model with 1/WC as dependent variable)” from the abstract, since it is more methodological issue and needs to be mentioned in the methodology only, together with logarithmic transformation. Those are procedures ordered to achieve normality of distribution required for regression analyses, and such information in the abstract is not necessary, and only can confuse the reader.
Methods:
lines 144-152: to which “meal patterns” this section is related? UPF/MPF or early/late eating? Please, specify
lines 279-283 (results): should go to methods
Results:
line 197: give p value with 3 decimals (uniformly in the text)
lines 206-210: define PF abbreviation (was it meant actually MPF and UPF instead of culinary ingredients and PF?); give units for energy: 10.7 (6.4- 14.1) and 5.2 (3.0-8.0) (which unit?); give p value with 3 decimals.
line 216: give some more text for table 5. Is this table 5 or 4? Now, it seems that eating sweets is negatively associated with WC, particularly in women, which is kind of strange finding, so please be sure that the regression methods were properly performed (see above)
line 278, 279 and 285 (and anywhere in the text where appropriate): please, specify “the early/medium/late meal patterns” or “the early/ late meal patterns.” (only saying “meal patterns” is too broad term, and can be related to any other type of meal patterns (e.g., UPF and MPF-meal pattern, as said in methods lines 147, frequent or not-frequent meal pattern, home or outside, etc.). Please, make clear distinction of “UPF/MPF meal patterns” with “early/late meal patterns” in the manuscript text.
lines 279-283: should go to methods (describing “early/late meal patterns” identiffication)
Discussion
lines 317, 318, 349 (and anywhere in the text where appropriate): instead of “UPF” please use ”UPF consumption” (also correct in other parts of manuscript where it is missing)
lines 344-347: are those lines deleted and why? In my opinion, the statement was good
line 359: what is meant by “the population characteristics.”? Are there any other explanations that UPF increased risk for high WC but not BMI in males? In men, sweets were the most associated with WC, probably some specific food can be more associated with centripetal obesity?
lines 359-360: “Interestingly, no single processed food was related to WC, suggesting a possible synergistic effect of foods or way of life in a “holistic diet” approach.” – Actually, the statement is not in accordance with the data from table 5: there was a significant negative association between sweets and WC, in total sample and particularly in women (which is kind of strange finding, isn’t it?). Please, make the stepwise regression models, with the 9 specific UPF, sex, age and other confounders included. (make this also for BMI, maybe there will be some significant associations?)
lines 361-362: are those lines deleted and why? In my opinion, the statement was good
lines 363-369: why here (in this study) there was not an increase in BMI with UPF? Was the energy intake balanced with the energy needs in UPF eaters, but probably macronutrient composition was not adequate (e.g., less protein and dietary fibers, minerals and vitamins, but more carbohydrates, sugars and fats which predispose for centripetal adiposity and insulin resistance? maybe some other components of UPF, such additives, syrups, molasses and oligosacharides, trans-fats, products from packaging like BPA can lead to insulin resistance and hormonal disturbances, leading to centripetal adiposity in men?). Are ghrelin and PYY and inflammatory markers more associated with centripetal than with general adiposity? Please, give some more considerations and explanations for no association with BMI, but more with centripetal adiposity (I have given here some ideas).
Conclusion
lines 441-443: “In conclusion, the present study focused on a particular subgroup in high risk of consuming UPF products, i.e. university students, and provided the first evidence regarding UPF consumption in young Greek adults.”- Please, rephrase into more simple, precise and straightforward statement: ” In conclusion, the present study confirmed a high UPF consumption among Greek university students”
line 449: “reductions in UPF consumption and a lower cardiovascular risk” – please, correct into “reductions in UPF consumption to lower centripetal adiposity, associated with increased cardiometabolic risk”
